# Effects of Caries Activity on Compositions of Mutans Streptococci in Saliva-Induced Biofilm Formed on Bracket Materials

**DOI:** 10.3390/ma13214764

**Published:** 2020-10-26

**Authors:** Bum-Soon Lim, Bo-Hyun Kim, Won-Jun Shon, Sug-Joon Ahn

**Affiliations:** 1Dental Research Institute and Department of Dental Biomaterial Science, School of Dentistry, Seoul National University, Seoul 03080, Korea; nowick@snu.ac.kr; 2Dental Research Institute and Department of Orthodontics, Seoul National University, Seoul 03080, Korea; holosugi89@naver.com; 3Dental Research Institute and Department of Conservative Dentistry, School of Dentistry, Seoul National University, Seoul 03080, Korea; endoson@snu.ac.kr

**Keywords:** caries activity, caries-active, caries-free, orthodontic bracket, mutans streptococci

## Abstract

This study aimed to investigate effects of caries activity on composition of mutans streptococci in saliva-induced biofilms formed on bracket materials. Three bracket materials were used as specimens: ceramic, metal, and plastic. After saliva was collected using a spitting method from caries-active (CA, decayed, missing, filled teeth (DMFT) score ≥ 10) and caries-free (CF, DMFT score = 0) subjects, saliva was mixed with growth media in a proportion of 1:10. The saliva solution was then incubated with each bracket material. After a saliva-induced biofilm was developed on the surface of the bracket material, the amounts of total bacteria and mutans streptococci were determined using real-time polymerase chain reaction. The results showed that biofilms from CA saliva contained more mutans streptococci but less total bacteria than biofilms from CF saliva, regardless of material type. Adhesion of total bacteria to ceramic was higher than to plastic, regardless of caries activity. Mutans streptococci adhered more to ceramic than to metal and plastic in both biofilms from CA and CF saliva, but there was a difference in adhesion between *Streptococcus mutans* and *Streptococcus sobrinus*. The amount of *S. mutans* was higher than that of *S. sobrinus* in biofilms from CA saliva despite similar amounts of the two strains in biofilms from CF saliva. The stronger adhesion of *S. mutans* to ceramic than to metal and plastic was more evident in biofilms from CA saliva than in biofilms from CF saliva. This study suggests that caries activity and material type significantly influenced composition of mutans streptococci in biofilms formed on bracket materials.

## 1. Introduction

Enamel decalcification around orthodontic appliances is the most common complication of orthodontic treatment with fixed appliances [1]. Prevalence of enamel decalcification in the orthodontic patients is about 12.6% to 50% per patients [1,2]. Placement of orthodontic brackets alters the oral environment, favoring enamel decalcification on the tooth surface, since it complicates mechanical cleaning around teeth surfaces and provides additional sites for biofilm formation [3]. Despite recent advancement in orthodontic materials and techniques as well as a variety of prevention and therapeutic methods, enamel decalcification during orthodontic treatment has not been overcome [1].

The enamel decalcification is induced by organic acids produced by oral bacteria. Among the various oral bacteria, mutans streptococci (mainly *Streptococcus mutans* and *Streptococcus sobrinus* in the human oral cavity) are the most important cariogenic bacteria for enamel decalcification due to their ability to form biofilm on the enamel surface and produce organic acid [4]. Several studies have shown increased level of mutans streptococci after bonding orthodontic brackets [5,6].

Oral biofilm is a complex microbial community that contains commensal and caries-related bacteria. When caries-related bacteria increase and commensals are reduced by ecological changes, such as placement of orthodontic brackets, biofilm may be transformed from a non-cariogenic to a cariogenic state, inducing enamel decalcification [7]. To find solutions for enamel decalcification, many studies have investigated biofilm formation on orthodontic materials using laboratory-based experimental biofilm models [8,9,10,11]. Mono-species biofilm models (mainly *S. mutans*) have been used, mainly for orthodontic purposes [8,10,11]. Although recent studies have investigated the relationships between biofilm formation and orthodontic issues using multi-species biofilm models [9,12], they used a defined consortium consisting of 10 to 12 bacterial species. Considering that the oral cavity is a heterogeneous environment consisting of more than 700 bacterial species [13], previous biofilm models could not properly reproduce oral biofilm-mediated diseases in vitro. To maintain the microbial complexity and heterogeneity of oral biofilms, the natural ecosystem should be applied to an experimental biofilm model. Since unstimulated whole saliva samples are recognized as a representation of the entire oral ecosystem [14,15], use of unstimulated whole saliva as a biofilm inoculum may be the best solution to simulate the complex oral environment.

In addition, significant compositional differences in biofilms around orthodontic brackets may exist between patients with low and high caries activities, because mutans streptococci are dominant in the oral cavity of subjects with high caries activity [16]. However, there has been no attempt to compare the composition of mutans streptococci in biofilms formed on orthodontic brackets with respect to caries activity. Information on the effects of caries activity on differences in composition of mutans streptococci in bracket biofilms will aid clinicians in managing orthodontic patients with high caries risk. Therefore, the aim of this study was to analyze the effects of caries activity on compositional differences in mutans streptococci in experimental biofilms formed on bracket materials. For this purpose, saliva-induced microcosm biofilms were created on each bracket material after saliva was collected from subjects with high caries activity (decayed, missing, filled teeth (DMFT) score ≥ 10) and low caries activity (DMFT score = 0) according to the criteria proposed by the World Health Organization (WHO) [17]. The null hypothesis was that there would be no significant difference in composition of mutans streptococci in saliva-induced microcosm biofilms formed on bracket materials.

## 2. Materials and Methods

### 2.1. Sample Preparation

We used three bracket materials as specimens: ceramic (Polycrystalline alumina, Hubit, Anyang, Korea), metal (stainless steel, Hubit, Anyang, Korea), and plastic (polycarbonate, Hubit, Anyang, Korea). Each material was provided in a uniform size (3.0 mm in thickness and 12.0 mm in diameter) from the manufacturers. A total of 33 bracket materials (11 specimens for each material, 10 for biofilm experiments and surface analysis, and one for SEM) was used.

### 2.2. Surface Analysis

Surface roughness (SR), surface wettability, and surface morphology were examined to analyze the associations between surface characteristics and biofilm composition. Confocal laser scanning microscopy (LSM 5 Pascal, Carl Zeiss MicroImaging GmbH, Göttingen, Germany) was used to calculate the average SR within the sampling area (450 × 450 × 50 μm). SR were measured at five random points of each specimen. The sessile drop method was used to analyze surface wettability by measuring water contact angle of the specimens as previously described [12,18]. The water contact angle was determined by an average between left and right contact angles of each drop using a contact angle analyzer (Phoenix 300, SEO, Suwon, Korea). The SR and water contact angle were evaluated from surfaces of all specimens prior to biofilm experiments. For evaluating surface morphology, one specimen was randomly selected from each group. The specimens were sonically cleaned in an isopropyl alcohol bath for 1 min and then gold coated. Each material was evaluated using SEM (S-4700 microscope, Hitachi, Tokyo, Japan) with a magnification set at ×500.

### 2.3. Subject Recruitment

Adult subjects were recruited after clinical oral examination by an experienced dentist. The inclusion criteria were (1) no antibiotic treatment in the past three months and (2) no systemic disease or abnormal tooth structure. The number of decayed, missing, and filled teeth was recorded according to the criteria proposed by WHO [17]. Because DMFT score ≥ 5 is considered as caries-active (CA), the subjects were divided into two groups according to caries activity: CA subjects (DMFT score ≥ 10 and more than one active carious lesion) and caries-free (CF) subjects (DMFT score = 0). Four CA subjects (two females and two males, age range 20–48 years, mean DMFT score = 11) and four CF subjects (one female and three males, age range 22–38 years) agreed to participate in this study (Table 1). The Institutional Review Board of the University approved the study protocol (S-D20170021).

### 2.4. Saliva Collection

For an inoculum for biofilm formation, unstimulated whole saliva (5 mL) was collected from each subject using a spitting method as previously described [12]. The subjects abstained from any toothbrushing, eating, or drinking at least 2 h before saliva collection. Saliva samples within groups were pooled (a total of 20 mL per group) and stored as aliquots of 1 mL at −80 °C until further analyses.

### 2.5. Biofilm Formation

A basal mucin medium (BMM, pH = 7.2) was used as a biofilm medium to simulate the saliva and supply nutrition sources as previously described [18]. It contained 2.5 g/L procine gastric mucin, 2 g/L proteose peptone, 2.5 g/L KCl, 1 g/L yeast extract, 1 g/L trypticase peptone, 0.1 g/L cysteine hydrochloride, 0.001 g/L hemin, 9 mM glucose, 1 mM sucrose, and 5 mM urea.

Biofilm assays were performed using polystyrene 24-well (flat-bottom) culture plate (SPL, Pocheon, South Korea). For biofilm assays, the stored saliva was thawed and mixed with BMM in a proportion of 1:10, and the solution was added to each well containing a bracket material. After incubation at 37 °C in an anaerobic jar (BD GasPak 150 Large Anaerobic System, Franklin Lakes, NJ, USA) for 48 h, the biofilm medium was removed, and the bracket materials were washed with 1.0 mL sterile phosphate-buffered saline (PBS, pH = 7.2) to remove unattached cells. After each material was transferred to a sterile tube containing 10 mL PBS, the biofilm was detached by sonication using four 30-s pulses at 25 W with three 30-s intermittent cooling stages as previously described [12]. The cell suspensions were centrifuged at 12,000 rpm for 15 min and washed with 1.0 mL sterile PBS.

### 2.6. Microbial Analysis

Microbial analyses were performed blindly using saliva samples of each caries group (baseline microbial composition) and biofilm suspension. An AccuPrep Genomic DNA Extraction kit (Bioneer, Daejeon, Korea) was used to extract bacterial DNA according to the manufacturer’s instructions. The PCR primers for of *S. mutans* and *S. sobrinus* were designed to amplify their dextranase genes [18]. The specific PCR primers for targeting a conserved region in the 16S rRNA gene were designed to measure the amount of total bacteria [18]. After DNAs were extracted from *S. mutans* ATCC 700610 and *S. sobrinus* ATCC 27607 for preparing the DNA standard curve, the number of bacteria in the samples was calculated from the standard curves. The CFX Connect system (Bio-Rad, Hercules, CA, USA) was used for real-time PCR. The PCR reaction mixtures contained 10 pM forward and reverse primers, 2 µL purified DNA, and 10 µL Accupower PCR 2× Green Star qPCR Master mix (Bioneer, Daejeon, Korea). Distilled water was added to a final volume of 20 µL. The primer sequences and PCR conditions are described in Table 2. CFX connect Software (Bio-Rad CFX Maestro, version 1.0, Bio-Rad, Hercules, CA, USA) was used to analyze PCR data. All the microbial analyzes were carried out in triplicate and independently repeated five times.

### 2.7. Statistical Analyses

Surface roughness (SR), water contact angle, salivary level of mutans streptococci, and level of mutans streptococci in biofilm were analyzed. After Kruskal–Wallis test was used to compare SR and water contact angle among bracket materials, Mann–Whitney U tests were used to analyze between-group differences. Salivary level of mutans streptococci between CA and CF subjects was analyzed using Mann–Whitney U test. Differences in amounts of mutans streptococci with respect to caries activity and material type were analyzed using two-way factorial analysis of variance (ANOVA) (IBM SPSS Statistics, version 24, IBM, Armonk, NY, USA). *p* values < 0.05 were considered statistically significant.

## 3. Results

Significant differences in SR and water contact angle were found between bracket materials. Ceramic exhibited higher SR than metal and plastic (ceramic > metal, plastic; Kruskal–Wallis test, *p* < 0.001), while plastic exhibited higher water contact angle than ceramic and metal (ceramic, metal < plastic; Kruskal–Wallis test, *p* < 0.001) (Table 3). Surface morphology according to SEM exhibited similar results to SR. Ceramic (Figure 1a) showed more irregular surfaces than metal (Figure 1b) and plastic (Figure 1c).

A significant difference in salivary microbial composition was found between the two caries groups. Although the amount of salivary total bacteria was similar between CA and CF subjects (Mann–Whitney U test, *p* > 0.05), salivary mutans streptococci were more prevalent in CA subjects than CF subjects (CA > CF; Mann–Whitney U test, *p* < 0.001) (Table 4). This indicates a dissimilarity in composition of mutans streptococci in the salivary microbiome between CA and CF subjects.

Table 5 demonstrates that caries activity significantly influenced the bacterial composition in biofilms formed on bracket materials. The amount of total bacteria was higher in CF biofilms than in CA biofilms (CA < CF; two-way ANOVA, *p* < 0.05), but the amount of mutans streptococci was higher in CA biofilms than in CF biofilms (CA > CF; two-way ANOVA, *p* < 0.001). There was also a significant difference in bacterial composition of bracket biofilms according to material type. The amount of total bacteria was higher in biofilms formed on ceramic than plastic, regardless of caries activity (ceramic > plastic; two-way ANOVA, *p* < 0.01). In *S. mutans*, however, a significant interaction was found between caries activity and material type (two-way ANOVA, *p* < 0.001). Although ceramic exhibited stronger adhesion of *S. mutans* than metal and plastic in both CA and CF biofilms, the differences in amount of *S. mutans* between ceramic and metal or plastic were higher in CA biofilms (ceramic >> metal, plastic; two-way ANOVA, *p* < 0.001) than in CF biofilms (ceramic > metal, plastic; two-way ANOVA, *p* < 0.05) (Table 5). *S. sobrinus* adhered more strongly to ceramic than to metal and plastic, regardless of caries activity (ceramic > metal, plastic; two-way ANOVA, *p* < 0.01). Interestingly, the amount of *S. mutans* was greater than that of *S. sobrinus* in CA biofilms despite the similar amounts between the two bacteria in CF biofilms.

## 4. Discussion

Enamel decalcification around orthodontic brackets has been of great concern to orthodontic patients and clinicians. Orthodontic brackets significantly contribute to enamel decalcification, because their complex design enhances biofilm formation by inhibiting access to the enamel surfaces for proper cleaning [1]. As a result, biofilm is developed adjacent to brackets even in patients with good oral hygiene. Preventing enamel decalcification is important to orthodontists, because these lesions are not esthetic and are potentially irreversible [19].

There are three bracket materials available in the market: stainless steel metal, ceramic, and plastic brackets. Metal brackets are the most popular, because these brackets have several practical and mechanical advantages, including size, strength, durability, and decreased frictional resistance [20]. However, metal brackets have esthetic disadvantages like metallic-like appearance. The other brackets became available due to patients’ demands for improved esthetics. Plastic brackets were introduced in the market as the esthetic alternative to metal brackets. The plastic brackets, while esthetically satisfactory in the early stages of treatment, deteriorated in appearance with time and were insufficiently strong to withstand long treatments or transmit torque [20]. Most ceramic materials used in orthodontic brackets are composed of aluminum oxide, either in its polycrystalline or monocrystalline form. The advantage of ceramic brackets over metal brackets is esthetics, but there are several disadvantages such as bracket fracture and tooth damage during bracket removal [20]. 

To determine a method to prevent enamel decalcification around orthodontic brackets, relationships between orthodontic brackets and biofilm formation have been extensively investigated using biofilm models with a single bacterial species or a bacterial consortium containing 10–12 laboratory strains [8,21,22]. However, these biofilm models cannot simulate the genuine oral environment because over 700 bacterial species exist in the oral cavity [13] and the cariogenic behavior of laboratory bacterial strains is not significantly different from that of clinical strains isolated from the human oral cavity [23]. Recently, a microcosm biofilm using human saliva as an inoculum has been introduced to more accurately mimic the complexity of real biofilm in vitro [24].

Attachment of orthodontic brackets to teeth surfaces may significantly influences microbial composition in the oral cavity, because a complex design of brackets facilitates bacterial adhesion and biofilm development by inhibiting proper cleansing around orthodontic brackets [3]. In addition, the microbial changes associated with orthodontic brackets may be significantly different between CA and CF subjects due to significant differences in intraoral microbial composition between them [16]. However, few studies have investigated the effects of caries activity on biofilm formation on bracket materials using microcosm biofilm models. In this study, we used a saliva-induced microcosm biofilm to compare differences in amount of mutans streptococci in the experimental biofilms formed on bracket materials.

This study demonstrated a significant difference in level of salivary mutans streptococci between CA and CF subjects (Table 4). The amounts of salivary mutans streptococci were higher in CA subjects than CF subjects despite similar amounts of salivary total bacteria between CA and CF subjects. It seems that homeostasis among microorganisms maintains the level of salivary total bacteria in CA subjects despite a larger proportion of salivary mutans streptococci. Considering that saliva is a reservoir for oral bacteria, higher amounts of salivary mutans streptococci in CA subjects may place them at high risk for enamel decalcification during orthodontic treatment.

When comparing bacterial composition in biofilms formed on bracket materials, caries activity and bracket materials significantly influenced biofilm composition (Table 5). The results demonstrated that CF biofilms had higher amounts of total bacteria than CA biofilms, while the amount of mutans streptococci was higher in CA biofilms than in CF biofilms. In particular, the level of *S. mutans* in CA biofilms was over 150 times higher than that of *S. mutans* in CF biofilms, irrespective of material type (Table 5). This may be because *S. mutans* is more prevalent in saliva from CA subjects than from CF subjects (Table 4). Higher levels of salivary mutans streptococci may increase their proportion in biofilms and produce more organic acid through fermenting carbohydrates, promoting selection and promotion of mutans streptococci proliferation in CA biofilms. As acidogenic mutans streptococci are selected and enriched, commensals previously prevalent in biofilms are decreased. As a result, the number of total bacteria may be decreased in CA biofilms.

In this study, the amount of *S. sobrinus* was lower than that of *S. mutans* in CA biofilms despite their similar amounts in CF biofilms (Table 5). This is because *S. sobrinus* is less prevalent than *S. mutans* in saliva from CA subjects, compared to saliva from CF subjects (Table 4), and *S. sobrinus* has a lower binding affinity to underlying materials than does *S. mutans* [25].

Biofilm composition was significantly influenced by material type (Table 5). This study demonstrated that mutans streptococci more strongly adhered to ceramic than to metal and plastic in both CA and CF biofilms despite a difference in adhesion between *S. mutans* and *S. sobrinus* (Table 5). The higher adhesion of mutans streptococci to ceramic could be due to differences in surface characteristics between the bracket materials. Bacterial adhesion, specifically initial adhesion of early colonizes such as *S. mutans* and *S. sobrinus* to underlying materials, is directly influenced by surface characteristics since initial adhesion of early colonizers is the first step for biofilm formation on underlying materials [26]. Considering that a rough surface enhances bacterial adhesion by increasing the binding sites and inhibiting detachment of bacterial colonies [18,22], higher SR for ceramic compared to metal and plastic (Table 3 and Figure 1) may explain the stronger adhesion of mutans streptococci to ceramic than to metal and plastic. In addition, higher wettability enhances bacterial adhesion and biofilm development on dental materials because it is related to surface free energy [27]. Since the contact angle is inversely related to surface wettability [9], lower water contact angles on ceramic than on plastic (Table 3) may partly explain stronger adhesion of mutans streptococci to ceramic than to plastic.

In this study, the adhesion pattern of total bacteria is similar to that of mutans streptococci. The amount of total bacteria was higher in biofilms on ceramic than those on plastic, irrespective of caries activity (Table 5). Because initial adhesion of early colonizers results in subsequent colonization of middle and/or late colonizers and leads to maturation of the entire biofilm [28], adhesion of total bacteria may be associated with that of early colonizers such as *S. mutans*.

Interestingly, the differences in amount of *S. mutans* between ceramic and metal or plastic were higher in CA biofilms than in CF biofilms (Table 5). The level of *S. mutans* in CF biofilms on ceramic was approximately 40% higher than that of *S. mutans* in CF biofilms on metal or plastic, but the level of *S. mutans* in CA biofilms on ceramic was approximately 300% higher than that of *S. mutans* in CA biofilms on metal or plastic (Table 5). This is because caries-active *S. mutans* strains are more competent in adhesion and biofilm formation than caries-inactive *S. mutans* strains [23]. In addition, rougher and wettable ceramic surfaces (Figure 1 and Table 3) may provide a more favorable environment for promoting adhesion and growth of caries-active *S. mutans*. The high level of *S. mutans* in CA biofilms, specifically on ceramic, may increase risk for enamel decalcification around orthodontic brackets. Considering that the amount of *S. mutans* in CA biofilms on ceramic is approximately 200 times higher than in CF biofilms on ceramic (Table 5), this study suggest that material type may be a critical factor in orthodontic patients with high caries activity rather than those with low caries activity. As a result, ceramic brackets may not be recommended in subjects with poor oral hygiene and/or high caries activity.

This study has limitations. Our results were obtained from in vitro experiments. Although a laboratory biofilm model simulates and reproduces cariogenic biofilms to a similar extent to those found in in situ or in vivo studies [24], in vitro experiments cannot accurately simulate a highly diverse and complex bacterial ecosystem in the oral cavity. In addition, the saliva sample was collected only from caries-active and caries-free subjects, which cannot represent oral microbiome of general orthodontic patients, because most patients have low to moderate caries activities. Furthermore, other experiments using periodontal pathogens should be conducted to solve gingival inflammation during orthodontic treatments, one of the major orthodontic complication. Future in situ studies using a split mouth design or in vivo experiments including a larger sample are needed to analyze the effects of caries activity on oral health of orthodontic patients.

## 5. Conclusions

This study investigated the effects of caries activity on bacterial composition in biofilms formed on bracket materials. The results showed that CA biofilms contained more mutans streptococci but less total bacteria than CF biofilms. Adhesion of total bacteria to ceramic was higher than to plastic, regardless of caries activity. Mutans streptococci adhered more strongly to ceramic than to metal and plastic in both CA and CF biofilms, but there was a difference between two mutans streptococci. The amount of *S. mutans* was higher than that of *S. sobrinus* in CA biofilms despite their similar amounts in CF biofilms. In addition, the stronger adhesion of *S. mutans* to ceramic than to metal or plastic was more evident in CA biofilms than in CF biofilms. Considering that cariogenic mutans streptococci were more highly present in biofilms formed on ceramic and high caries activity further enhanced adhesion of mutans streptococci to ceramic, this in vitro study suggests that ceramic brackets should be carefully applied to orthodontic patients with high caries activity.

## Figures and Tables

**Figure 1 materials-13-04764-f001:**
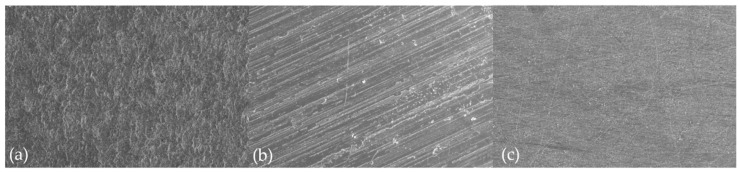
Scanning electron microscopic images of bracket materials at 500×: (**a**) ceramic (polycrystalline alumina), (**b**) metal (stainless steel), and (**c**) plastic (polycarbonate).

**Table 1 materials-13-04764-t001:** Information on caries activity of participants in this study.

Group		Age	Sex	Number of Decayed Teeth	Number of Missing Teeth	Number of Filled Teeth	DMFT Score
Caries-active (CA)	1	29	Male	3	0	7	10
	2	48	Female	2	0	9	11
	3	20	Female	5	2	6	13
	4	29	Male	3	0	7	10
Caries-free (CF)	5	22	Male	0	0	0	0
	6	31	Male	0	0	0	0
	7	38	Female	0	0	0	0
	8	33	Male	0	0	0	0

**Table 2 materials-13-04764-t002:** Primer sequences and cycling conditions used in this study.

Primer	Sequence (5’-to-3’)	Size of Amplicon (Base Pairs)	Initial Denaturation	Denaturation	Annealing	Extension	Cycles
Total Bacteria	Forward: TGGAGCATGTGGTTAATTCGA	160	94 °C	95 °C	60 °C	60 °C	40
	Reverse: TGCGGACTTAACCCAACA		30 s	20 s	45 s	10 s	
*Streptococcus mutans*	Forward: CTACACTTTCGGGTGGCTTG	261	94 °C	95 °C	60 °C	60 °C	40
	Reverse: GAAGCTTTTCACCATTAGAAGCTG		30 s	20 s	45 s	10 s	
*Streptococcus sobrinus*	Forward: AAAACATTGGGTTACGATTGCG	156	94 °C	95 °C	60 °C	60 °C	40
	Reverse: CGTCATTGGTAGTAGCCTGA		30 s	20 s	45 s	10 s	

**Table 3 materials-13-04764-t003:** Differences in surface roughness and water contact angle among bracket materials.

Surface Analyses	Ceramic	Metal	Plastic	Significance ^†^	*p* Value
Surface Roughness	0.41 ± 0.09	0.19 ± 0.08	0.27 ± 0.06	Ceramic > Metal, Plastic	0.000
Water Contact Angle	62.71 ± 6.08	65.98 ± 5.75	74.83 ± 6.09	Ceramic, Metal < Plastic	0.000

^†^ Kruskal–Wallis test was performed to compare differences in amounts of bacteria among bracket materials.

**Table 4 materials-13-04764-t004:** Bacterial composition in saliva collected from caries-active and caries-free subjects.

Counts per mL	Caries-Active	Caries-Free	Significance ^†^	*p* Value
Total Bacteria	1.0 × 10^8^ ± 0.1 × 10^8^	1.1 × 10^8^ ± 0.2 × 10^8^	Caries-Active = Caries-Free	0.745
*Streptococcus mutans*	6430.9 ± 1061.5	802.2 ± 153.6	Caries-Active > Caries-Free	0.000
*Streptococcus sobrinus*	97.8 ± 13.6	13.4 ± 5.9	Caries-Active > Caries-Free	0.003

^†^ Mann–Whitney U test was performed to compare differences in amounts of salivary bacteria between caries-active and caries-free subjects.

**Table 5 materials-13-04764-t005:** Bacterial composition in saliva-induced biofilms formed on ceramic, metal, and plastic materials with respect to caries activity.

Count per Specimen	Ceramic	Metal	Plastic	Significance ^†^	*p* Value
Total Bacteria				Caries-Active < Caries-Free Ceramic > Plastic	0.019 0.007
Caries-Active	2.4 × 10^7^ ± 1.0 × 10^7^	1.8 × 10^7^ ± 0.9 × 10^7^	1.3 × 10^7^ ± 0.5 × 10^7^		
Caries-Free	3.5 × 10^7^ ± 2.4 × 10^7^	2.1 × 10^7^ ± 1.5 × 10^7^	2.2 × 10^7^ ± 1.4 × 10^7^		
*Streptococcus mutans*				Caries-Active > Caries-Free	0.000
Caries-Active	1674.8 ± 482.2	560.8 ± 432.0	564.2 ± 478.9	Caries-Active (Ceramic >> Metal, Plastic)	0.000
Caries-Free	47.7 ± 19.3	34.1 ± 10.0	36.8 ± 13.2	Caries-Free (Ceramic > Metal, Plastic)	0.032
*Streptococcus sobrinus*					
Caries-Active	115.7 ± 48.6	91.4 ± 41.7	80.7 ± 43.3	Caries-Active > Caries-Free	0.000
Caries-Free	48.9 ± 21.9	28.8 ± 14.0	39.8 ± 16.8	Ceramic > Metal, Plastic	0.041

^†^ Two-way factorial analysis of variance was performed to compare differences in amounts of bacteria.

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
