# Peer review of "Effects of Caries Activity on Compositions of Mutans Streptococci in Saliva-Induced Biofilm Formed on Bracket Materials"

_materials, 2020, doi:10.3390/ma13214764_

Round 1

Reviewer 1 Report

Dear Sirs

Dear Sirs,

Authors aimed to study three bracket materials and how they effect on caries, the amounts of cariogenic bacteria in saliva and the surfaces. The study is interesting, but a lot of conclusions are made without clarifications and definition. The manuscript requires major revision including detail information and explanations what are behind each statement. I also recommend the English proof reading.

Major comments:

Comment1, line 2-4. Please state more clearly what was aims of the study and define ‘effects of caries activity on composition of cariogenic bacteria’ . Please define caries activity and cariogenic bacteria? Please re-write the title according to the corrections.

Comment 2, line 19 Please state more clearly and define ‘saliva-induced biofilm’ and how it was collected

comment 3, line 19. Please state more clearly and define (‘each surface ‘) which surfaces in teeth/mouth/etc .

comment 4 line 21. ‘CA biofilms’. Do you mean saliva or biofilm

Comment 5, line 22 and materials and methods. How the adhesion was defined? please tell

Comment 6, line 23 and lines 33-63. Please define how the caries activity was defined in the abstract and in intro

comment 7, line 24-25. Please define ‘but there was a difference between

 Streptococcus mutans and Streptococcus sobrinus’. How the difference occurred?

comment 8, line 83-91. please list all samples which were collected from the subjects

Comment 9. How Sem was performed? Please add it to the materials and methods part.

comment 10. Clinical features: age, sex, DMFT score, number of decayed, missing, and filled teeth etc. should be listed and reported.

comment 11 results: the used statistical methods should be reported after each p-values

comment 12. please discuss if or how different manufacturer’s material vary and clinician should select materials (what to consider with different patients).

Minor comment

Please use the English proof reading.

Author Response

Q1. line 2-4. Please state more clearly what was aims of the study and define ‘effects of caries activity on composition of cariogenic bacteria’ . Please define caries activity and cariogenic bacteria? Please re-write the title according to the corrections.

A1. Thank for the comment. Caries activity refers to the increment of active caries lesions, including new and recurrent lesions that occur over a stated period of time. In general, caries activity is assessed according to the WHO criteria and DMFS (decayed, missing, filled teeth) score ≥ 5 is considered as caries-active. In this study, the subjects were divided into two groups (caries active and caries free) according to the WHO criteria. I added information on caries activity to the Abstract (lines 18 – 20) and Introduction (lines 73 - 76).

Among the various oral bacteria, mutans streptococci (mainly Streptococcus mutans and Streptococcus sobrinus in the human oral cavity) are the most important cariogenic bacteria for enamel demineralization due to their ability to adhere to the tooth surface, produce organic acid, and endure under an acidic environment.

I changed the term ‘cariogenic bacteria’ to ‘mutans streptococci’, to avoid confusion as suggested by the first reviewer in the Title (lines 2 – 4). In addition, an explanation about mutans streptococci was added (lines 45 – 47) and the term ‘cariogenic bacteria’ was changed to ‘mutans streptococci’ throughout the manuscript.

Q2. line 19 Please state more clearly and define ‘saliva-induced biofilm’ and how it was collected

A2. I stated more clearly the method section in the Abstract after defining ‘saliva-induced biofilm’ and how it was collected (lines 18 – 22).

Q3. line 19. Please state more clearly and define (‘each surface ‘) which surfaces in teeth/mouth/et

A3. I changed the term ‘each surface’ to ‘the surface of each bracket material’ as suggested by the reviewer (line 22).

Q4. line 21. ‘CA biofilms’. Do you mean saliva or biofilm

A4. CA biofilms was the biofilms derived from CA saliva. I changed ‘CA biofilms’ and ‘CF biofilms’ to ‘biofilms from CA saliva’ and ‘biofilms from CF saliva’, respectively in the Abstract to avoid confusion as suggested by the reviewer (lines 23 – 31).

Q5. line 22 and materials and methods. How the adhesion was defined? please tell

A5. Biofilm formation is a serial event of bacterial adhesion. When oral biofilm begins, early colonizers are adhered first, then followed by adhesion colonization of the middle and late colonizers, resulting in maturation of the entire biofilm. When describing specific bacterial species in biofilms (for example, Streptococcus mutans, Streptococcus sobrinus, and total bacteria in this study), the term ‘adhesion’ is generally used (references 12 and 18).

Q6. line 23 and lines 33-63. Please define how the caries activity was defined in the abstract and in intro

A6. As described above, caries activity is assessed according to the WHO criteria and DMFS (decayed, missing, filled surface) score ≥ 5 is generally considered as caries-active. I added this information to the Abstract (lines 19 – 21), Introduction (lines 72 – 76), and Materials and Methods (line 102) as suggested by the reviewer.

Q7. line 24-25. Please define ‘but there was a difference between

A7. I changed the sentence ‘but there was a difference between’ to ‘but there was a difference in adhesion between’ as suggested by the reviewer (line 27).

Q8. line 83-91. please list all samples which were collected from the subjects

A8. I added detailed information on participants to the Table 1 as suggested by the reviewer (line 109).

Table 1. Information on caries activity of participants in this study

Group

Age

Sex

Number of decayed teeth

Number of missing teeth

Number of filled teeth

DMFT score

Caries active

1

29

Male

3

0

7

10

2

48

Female

2

0

9

11

3

20

Female

5

2

6

13

4

29

Male

3

0

7

10

Caries free

5

22

Male

0

0

0

0

6

31

Male

0

0

0

0

7

38

Female

0

0

0

0

8

33

Male

0

0

0

0

Q9. How Sem was performed? Please add it to the materials and methods part.

A9. I added detailed information to the Materials and Methods section as suggested by the reviewer (lines 94 - 96).

Q10. Clinical features: age, sex, DMFT score, number of decayed, missing, and filled teeth etc. should be listed and reported.

A10. I added detailed information on participants to the Table 1 as suggested by the reviewer (line 109).

Table 1. Information on caries activity of participants in this study

Group

Age

Sex

Number of decayed teeth

Number of missing teeth

Number of filled teeth

DMFT score

Caries active

1

29

Male

3

0

7

10

2

48

Female

2

0

9

11

3

20

Female

5

2

6

13

4

29

Male

3

0

7

10

Caries free

5

22

Male

0

0

0

0

6

31

Male

0

0

0

0

7

38

Female

0

0

0

0

8

33

Male

0

0

0

0

Q11. the used statistical methods should be reported after each p-values

A11. I added information on the used statistical methods in the Results as suggested (lines 156 – 190).

Q12. please discuss if or how different manufacturer’s material vary and clinician should select materials (what to consider with different patients).

A12. I added detailed information on bracket materials to the Discussion as suggested by the reviewer (lines 202 – 212).

Q13. Please use the English proof reading.

A13. The manuscript was revised by a professional English editing company.

Reviewer 2 Report

This paper analysis the effects of caries activity of two different saliva-induced biofilm (one from patients with active caries activity and the other from patients with no caries activity) on three different bracket materials: plastic, metal and ceramic. The paper is well organized but it has different flows. Please revise the whole manuscript as it has different typos. 

Introduction: please describe better the state of the art and the reason or the necessity of your work. There is lack of literature on this subject. To the reviewer knowledge this subject is very studied in dentistry and mostly in orthodontics. Thus, please better explain the reason of your work.  

  Materials and methods:  you state in the manuscript that: A total of 33 bracket materials (11 specimens for each material, 10 for biofilm experiments and surface 70 analysis, and one for scanning electron microscopy [SEM]) was used.However, you have performed other analysis like: Surface roughness, surface wettability, and surface morphology. It is not clear in this section how the specimens were divided. please specify.  Conclusions: please discuss better this section and moreover the limits of the study. it is reported in a skimpy way in the text.

Author Response

Comments on the suggestions by the second reviewer

Q1. Introduction: please describe better the state of the art and the reason or the necessity of your work. There is lack of literature on this subject. To the reviewer knowledge this subject is very studied in dentistry and mostly in orthodontics. Thus, please better explain the reason of your work.  

A1. I added more information on the reason or the necessity of this work in the Introduction (lines 38 – 49) and Discussion (lines 196 – 201) as suggested.

Q2. Materials and methods:  you state in the manuscript that: A total of 33 bracket materials (11 specimens for each material, 10 for biofilm experiments and surface 70 analysis, and one for scanning electron microscopy [SEM]) was used. However, you have performed other analysis like: Surface roughness, surface wettability, and surface morphology. It is not clear in this section how the specimens were divided. please specify. 
A2. In order to investigate relationships between surface characteristics and biofilm composition, both surface characteristics and biofilm formation should be conducted on the same material. Therefore, the surface roughness and water contact angle were evaluated from surfaces of all specimens prior to biofilm experiments. Because disc specimen could not be re-used after scanning electron microscopy (SEM) analysis, one specimen was randomly selected from each group and the specimen was used for SEM. Therefore, 10 bracket materials were used for both biofilm experiments and surface analysis, and one was used for SEM (a total of 11 specimens). This information was added to the lines 93 – 96.

Q3. Conclusions: please discuss better this section and moreover the limits of the study. it is reported in a skimpy way in the text.

A3. I added information on limitation of this study and re-wrote Conclusions as suggested by the reviewer (lines 305 – 308).

Reviewer 3 Report

Biofilms possess a critical challenge in infections and its related secondary infections. This article focuses on critical aspects of its growth and treatment options in oral cavity. 

It is an interesting article and is helpful for readers of various field. 

Author Response

Biofilms possess a critical challenge in infections and its related secondary infections. This article focuses on critical aspects of its growth and treatment options in oral cavity. It is an interesting article and is helpful for readers of various field. 

Thank you very much for the comments

Reviewer 4 Report

The topic under investigation is relevant and contains valuable information, and could be a good addition to the scientific literature. My following comments are suggested:

- Introduction is short, it should be extended.

- A better elaborated objectives paragraph should be reported. More detailed aims of the study would be advisable (e.g., PI/ECO(S,T) framework). In addition, authors could also report their pre-specified hypotheses.

- Abbreviations should not be abused. Concepts little used in text (e.g., SW) should be avoided or explained in an abbreviations paragraph.

- All variables analyzed should first listed, classified according to their nature (e.g., categorical, continuous…) and then describe tests used (variable by variable or groups).

- The exact p-values obtained should be stated in tables, not only p<0.001 and never “NS” (a clear example of selective reporting bias).

- Discussion, as introduction section should also be extended and potential biological mechanisms better discussed.

- Although the inclusion of a limitations paragraph is appropriate, it should be better elaborated. Authors should be more auto critical. The in vitro study is a design limitation, but more specific methodological problems should be discussed. For example, the experiments were not blinded (this good practice is seldom performed in these type of studies). Please, three or four limitations are welcome and would improve the transparency and integrity of this study.

- Finally, a paragraph including more precise recommendations for future studies would be highly recommended.

Author Response

Q1. Introduction is short, it should be extended.

A1. I added more information on the reason or the necessity of this work in the Introduction (lines 37 – 49) and Discussion (lines 196 – 201) as suggested.

Q2. A better elaborated objectives paragraph should be reported. More detailed aims of the study would be advisable (e.g., PI/ECO(S,T) framework). In addition, authors could also report their pre-specified hypotheses.

A2. I added a prespecified hypothesis and re-wrote the objective paragraph in the Introduction as suggested by the reviewer (lines 71 – 77).

Q3. Abbreviations should not be abused. Concepts little used in text (e.g., SW) should be avoided or explained in an abbreviations paragraph.

A3. I deleted abbreviation ‘SW’ in the text as suggested.

Q4. All variables analyzed should first listed, classified according to their nature (e.g., categorical, continuous…) and then describe tests used (variable by variable or groups).

A4. I re-wrote the statistical analyses section as suggested by the reviewer (lines 148 – 153).

Q5. The exact p-values obtained should be stated in tables, not only p<0.001 and never “NS” (a clear example of selective reporting bias).

A5. I added the exact p-values in the Tables as suggested (Tables 3, 4, and 5).

Q6. Discussion, as introduction section should also be extended and potential biological mechanisms better discussed.

A6. I added information on potential mechanisms of enamel decalcification to the Discussion as suggested (lines 196 – 201).

Q7. Although the inclusion of a limitations paragraph is appropriate, it should be better elaborated. Authors should be more auto critical. The in vitro study is a design limitation, but more specific methodological problems should be discussed. For example, the experiments were not blinded (this good practice is seldom performed in these type of studies). Please, three or four limitations are welcome and would improve the transparency and integrity of this study.

A7. The microbial experiments were performed blindly without any clinical information on caries activity. I added this information to the Materials and Methods section (line 131). In addition, experimental limitations were added to the Discussion as suggested by the reviewer (lines 290 – 295).

Q8. Finally, a paragraph including more precise recommendations for future studies would be highly recommended.

A8. I changed the paragraph containing the limitation and future recommendations as suggested by the reviewer (lines 290 – 295).

Reviewer 5 Report

Dear editor,

the research was done and described well. Results are important for clinical work.

Author Response

(The authors gave the same response as above.)
